# Providers perspectives on a team-based maternal health care delivery in Ghana: A qualitative study

Veronica Millicent Dzomeku[1], Ebenezer Dassah[2], Ebenezer Mensah Gyimah[2], Abigael Omowumi Emikpe[1], Lydia Boampong Owusu[1], Catherine Kroamah Dwumfour[1], Oluwatoyin Abayomi Ogunyewo[3], Thomas Peprah Agyekum[2], Eric Adjei Boadu[2], Emmanuel Kweku Nakua[2]*

1 School of Nursing and Midwifery, Kwame Nkrumah University of Science and Technology, Kumasi, Ghana, 2 School of Public Health, Kwame Nkrumah University of Science and Technology, Kumasi, Ghana, 3 Department of Nursing Science, University of Jos, Jos, Nigeria

* enakua.chs@knust.edu.gh

## Abstract

Interprofessional team-based care is crucial in ensuring respectful and dignified maternal services. However, there is limited research that explores this issue from the perspectives of health care providers in Ghana. The study sought to explore the perspectives and experiences of health care providers working in a collaborative team-based setting at a health in Ghana, with the aim to improve respectful and dignified maternal care. We used a descriptive qualitative study and conducted semi-structured interviews with 35 participants from diverse cadre of health care providers, including Midwives, Nurses, Nutritionists, Pharmacists, Physicians, Radiographers and Sonographers. The data were subsequently transcribed and analysed thematically. The findings revealed four overarching themes namely roles, facilitators, barriers and strategies to overcome barriers experienced by healthcare professionals within an interprofessional team-based setting providing respectful and dignified maternal services. Providers' primary roles in enhancing respectful and dignified maternal services within interprofessional team-based setting included enhancing patients' outcome, improving healthcare professionals' outcomes and optimizing facility outcomes. The facilitators to respectful and dignified maternal care were clear role definitions, transparent communication, personal empathy and professional competence. Conversely, barriers to the delivery of respectful and dignified maternal services within a team-based setting noted in participants' responses included infrastructural deficiencies, inadequate staffing, discrimination and negative professional attitudes. Participants' proposed strategies to overcome these barriers included investments in infrastructure, enhanced professional competence among staff and improved interprofessional communication within and between teams. Addressing these barriers could inform the development of policies and clinical practices aimed at advancing respectful and dignified maternal care. The study can also contribute to

---

**Data availability statement:** All relevant data are within the paper.

**Funding:** This study was supported by grant 1R25TW011217 from the US National Institutes of Health (NIH)/Fogarty International Center (FIC) which also includes co-funds from the U.S. Department of State's Office of the U.S. Global AIDS Coordinator and Health Diplomacy (S/GAC) and the President's Emergency Plan for AIDS Relief (PEPFAR) to the African Forum for Research and Education in Health (AFREhealth). The content is solely the authors' responsibility and does not necessarily represent the official views of the funders. The funders had no role in study design, data collection and analysis, decision to publish, or preparation of the manuscript.

**Competing interests:** The authors have declared that no competing interests exist.

the design and formulation of an operational manual required to shape interprofessional team-based respectful and dignified maternal care in Ghana and other similar contexts.

## Introduction

The World Health Organisation advocates for access to respectful and dignified maternal health services [1,2]. It is evident that respectful and dignified maternal health services reduce maternal mortality and morbidity rates, however, these care processes may be attainable when services are well-structured and organized [3,4]. To deliver respectful and dignified maternal healthcare services, Interprofessional Team-Based Care (IPTBC) is crucial in ensuring respectful and dignified maternal services. Given this, scholars have espoused the need for collaboration among healthcare professionals in order to enhance access to respectful and dignified maternal healthcare [5]. Meanwhile IPTBC has been bedeviled with provider dissatisfaction and poor communication among healthcare workers emanating from negative attitudes toward other health professionals [6]. As such, most women experience fragmented, uncoordinated, and inconsistent maternal services marked by disrespect, especially in low- and middle-income countries [4,7–10]. Others also experienced traumas emanating from negative attitudes from healthcare workers, discriminations, poor psychosocial support and overly medicalized procedures among other human rights abuses [7,11]. As such, interprofessional team-based respectful and dignified maternal care is imperative to achieving optimal healthcare outcomes for all women [12–15] and their future healthcare seeking behaviours [16–18].

Interprofessional care (IPC) is increasingly recognized for its capacity to enhance collaboration among health professionals and improve patient outcomes [19]. A key driver of IPC is interprofessional education (IPE), which trains healthcare providers from diverse fields to work effectively as a team [20]. Initiatives such as the STRIPE HIV program have demonstrated the efficacy of IPE in improving knowledge, teamwork, and collaborative attitudes among healthcare professionals, with allied health students in Ghana showing strong readiness for IPE [21]. However, barriers such as professional silos, traditional discipline-specific training systems, inadequate infrastructure, and cultural attitudes undermine the effectiveness of IPC in Ghana [22]. These challenges necessitate targeted reforms in policies, infrastructure, and educational frameworks to bridge gaps and foster teamwork. Despite these limitations, successes in regions such as the Ashanti highlight IPC's potential to address local health needs and improve outcomes through tailored initiatives [22]. The predominantly young workforce, with many under 30 and engaged in programs like STRIPE HIV, underscores the importance of embedding IPC principles early in professional education to cultivate collaboration [23]. In maternal healthcare, IPC holds transformative potential to advance respectful and dignified care by fostering collaboration, cultural sensitivity, and accountability. Addressing systemic challenges is crucial for minimizing disrespect and abuse during maternity care optimizing the impact of IPC,

enabling healthcare systems to improve maternal health outcomes and equity through integrated and culturally sensitive care practices.

Globally, it is evident that disrespect and abuse during maternity care hinder facility-based deliveries, crucial for reducing maternal and neonatal mortality [24,25]. For instance, most women experience verbal, physical, and psychological abuse, often linked to systemic inadequacies, provider biases, and structural barriers to respectful and dignified maternal care. In Ghana, research on disrespect and abusive maternal care highlights similar systemic challenges in maternal healthcare delivery. Extant literature has focused on patient-reported experiences [26,27], and also explored midwives' perspectives [28,29], identifying structural barriers such as workload and institutional protocols[30]. During access to maternal healthcare services, women in Ghana have reportedly experienced disrespect and abuse in the form of physical, verbal, psychological abuse, neglect, and confidentiality violations[31,32]. There are evidence of discrimination against poor, young, or HIV-positive mothers [33,34]. Also, victim-blaming attitudes persist among providers, who rationalize disrespect and abuse due to systemic constraints[30]. Significantly, disadvantaged and teenage mothers face higher risks of disrespect and abuse in seeking access to maternal care, with midwives, aged 31–48, with an average of eight years' experience, reportedly speaking about these challenges [29]. Addressing disrespect and abuse in accessing maternal care requires healthcare reforms, training on respectful and dignified maternity care, and alignment with relevant local Patient Charter [29,35] and international maternal human right frameworks such as the Respectful Maternity Care (RMC) Charter developed by the White Ribbon Alliance [36]. Empowering women through public education and policy enforcement is also vital. Holistic interventions targeting systemic, individual, and societal drivers can improve respectful and dignified maternity care.

Respectful and dignified maternal care has gained increasing attention globally as a concept for enhancing maternal health outcomes and addressing systemic inequities [37]. It emphasizes a rights-based approach to maternity care that upholds dignity, privacy, autonomy, and respect for women during childbirth. Research trends indicate a shift towards integrating respectful and dignified maternal care into broader maternal health policies and institutional practices, particularly in low- and middle-income countries (LMICs)[38,39]. Recent studies have focused on understanding the barriers to respectful and dignified maternal care and implementing culturally tailored interventions to promote its adoption [38,40]. Respectful and dignified maternal care is vital for enhancing the quality, accessibility, and outcomes of maternity care globally. As per the narrative proposed by the WHO, a care that ensures dignity, informed decision-making, and continuous support during childbirth, fostering positive maternal experiences and minimizing mortality describes respectful and dignified maternal care [1]. However, systemic barriers such as resource constraints, gender discrimination, and cultural norms contribute to disrespectful practices, particularly in low- and middle-income countries (LMICs), where income disparities exacerbate mistreatment[41,42]. In Ghana, logistical challenges like limited infrastructure for privacy, understaffing, and inadequate training hinder midwives' ability to deliver respectful care, while heavy workloads and negative attitudes compound the problem [37]. Programs like the White Ribbon Alliance's Charter and WHO guidelines emphasize training in cultural competence and communication, addressing systemic deficiencies to embed respectful practices into healthcare frameworks[35]. Respectful care not only builds trust in healthcare systems and encourages health-seeking behavior but also significantly improves maternal and neonatal outcomes. Conversely, disrespect deters service use, increasing risks of complications and mortality [43,44]. Younger women and those of lower socioeconomic status are particularly vulnerable to disrespectful treatment [45]. To overcome these challenges, comprehensive reforms in policy, infrastructure investment, and healthcare training are essential. Integrating respectful maternal care into healthcare systems is transformative, promising equity, satisfaction, and better health outcomes, particularly for underserved populations. Further research and investment are critical to scaling these practices in LMICs[35,39].

Nonetheless, in Ghana, significant strides have been made through policy commitments to address the limited interprofessional team-based respectful and dignified maternal healthcare. Government-led initiatives (i.e., campaigns for skilled birth assistance, safe motherhood initiative, prevention of mother to child transmission for HIV, the promotion of

facility-based childbirths, folic acid supplementation and free maternal healthcare services), have not only minimized maternal mortality, but significantly encouraged collaborations among diverse healthcare professional [46], thereby enhancing access to respectful and dignified maternity care.

Given the factors contributing to limited interprofessional team-based respectful and dignified maternal healthcare, it is crucial to devise and implement strategies aimed at augmenting access to high-quality comprehensive multidisciplinary services for this unique population [47]. This necessitates the need for collaborative efforts among professionals towards enhanced respectful and dignified maternal care. Team-based interprofessional collaboration could address the limited number and cadre of healthcare providers in the existing maternity care system and ensure enhanced respectful and dignified maternity care services [48]. This approach could also deal with both the complexities underlying access to respectful and dignified maternal healthcare and also prioritize targeted interventions among women accessing antenatal, intrapartum and postpartum care [49], particularly those in low- and middle-income countries (LMICs).

In Ghana, existing studies have explored the provision of culturally congruent care by interprofessional teams within a pediatric setting [50] and interprofessional collaboration in HIV care [22]. However, there has been limited evidence on the perspectives and experiences of various cadres of healthcare workers providing respectful and dignified maternal healthcare services within an interprofessional team-based collaborative hospital setting. Therefore, we aimed to investigate the perspectives and experiences of healthcare workers on interprofessional team-based respectful and dignified maternal care in Ghana. The findings of this study could help inform the development of specific and actionable research, policy and clinical practice interventions to improve interprofessional team-based collaborative respectful and dignified maternal care.

## Method

### Research design

We employed an exploratory descriptive qualitative design to explore and describe the experiences and perspectives of healthcare professionals on interprofessional team-based respectful and dignified patient care. This approach is appropriate as it allows researchers to develop a comprehensive understanding of participants' experiences of their practice in a real-world context [51–53]. The study was conducted from February to April 2024.

### Study setting

We carried out the study in a secondary level health facility in the Ashanti Region of Ghana. The health facility has a 140-bed capacity providing healthcare services for almost one-third (30.3%) of the residents in Kumasi. The facility is classified as secondary level because it serves as a major referral point for many primary-level health facilities, and also renders services such as general medical care, public health, diagnostics, training of medical residents and specialist services (i.e., Maternal Care, Herbal Clinic, Ear Nose and Throat, Gynecology and Obstetrics, Ophthalmology and Pediatrics). The facility has a Maternity Unit with different cadres of health professionals providing team-based care to clients.

### Participants recruitment and sampling

Participants for the study included Midwives, Nurses, Nutritionists, Pharmacists, Physicians, Radiographers and Sonographers. We recruited these participants purposively, striving for diversity in representation of age, gender, type of health professional, years of work experience and unit/department. Inclusion criteria were being health professional with at least one year working experience and willing to participate in the study. We excluded health professionals on internship/rotation and those who performed only administrative duties. We identified and recruited participants through the hospital administration. The hospital administrator wrote an introductory letter stating the study purpose. Three members of the research team then contacted potential participants with the study information and consent form. The sample size for this study was guided by the concept of "information power," where adequate sample size is defined in terms of a clearly defined aim, specific sample, theoretical approach, high quality dialogue and clear analytic strategy [54].

## Data collection

We developed an interview guide based on the objectives of the study and extensive literature on the topic [4–7,22,55]. We pilot-tested the guide with three participants who were not part of the main study. The interview guide encompassed health professional roles, facilitators, challenges and how to improve IPTBC in relation to dignified and respectful care. Two trained research assistants conducted face-to-face semi structured interviews in English. All interviews were conducted at a time and place in the hospital that was convenient for the participants and the research assistants. Participants demographic data were collected at commencement of the interview. All interviews were audio recorded, with permission from participants, and ranged from 45 and 60 minutes.

## Data management and analysis

The audio recordings were transcribed verbatim by research assistants prior to the data analysis. The interview transcripts were checked by the research team against the respective audios for quality control purposes. The completed transcripts were then imported into NVivo12, a qualitative data management software, for coding and analysis.

We followed the thematic analysis procedures outlined by [56] to analyze the data. Three researchers (VMD, ED and EMG) read and re-read the interview transcriptions to familiarize with the data. The researchers then discussed and agreed on the unique codes and categorized them to obtain major themes. The themes were then further developed and defined by giving them names and concise working definitions that captured the essence of each theme as it relates to the objective of the study. Major themes and patterns were then discovered in the data, thus providing insights into the experiences of the participants. The major themes and subthemes were finally defined and named by the research team.

## Rigor

We developed several strategies to enhance rigor or data trustworthiness, and these include credibility, transferability, dependability and confirmability [57,58].To enhance credibility, we reviewed, discussed and reached consensus on the themes and sub-themes. We also reported the participants' responses using verbatim quotations. We presented the findings with a clear description of settings, participants, methods and the process of data analysis to enhance transferability. Further, we achieved dependability by detailing the adopted research process to allow other researchers to replicate our findings. Confirmability in this study was ensured using audit trials, by which we carefully documented our beliefs and decisions made throughout the data collection and analysis processes.

## Ethical considerations

The Kwame Nkrumah University of Science and Technology Committee for Human Research and Ethics (Reference– CHRPE/AP/1095/23) provided ethical clearance for the conduct of the study. In addition, we obtained approval from the health facility where we conducted the study (KSH/RESH-50). All study procedures were conducted in line with related guidelines and regulations. Written informed consent was obtained from participants prior to participation in the interviews. Participant codes were used to ensure anonymity and confidentiality.

## Findings

**Demographic characteristics of participants.** Thirty-five (35) participants were included in the study (see Table 1 for details of participants' demographics). The mean age of the participants was 32 years with a range from 23 to 45 years. In terms of gender, participants consist of 28 (80%) females and 7 (20%) males. Most of the participants were midwives (n = 19, 54%). The overall mean work experience and work experience were 6.5 years and 5.4 years respectively.

**Table 1. Socio-Demographic Characteristics of Participants.**

| Characteristic | Number (%) of participants (n = 35) |
|---|---|
| **Age in years (mean ± SD)** | 32.3 ± 5.8 |
| **Gender** | |
| Female | 28 (80.0) |
| Male | 7 (20.0) |
| **Cadre of health care provider** | |
| Nurses | 2 (5.7) |
| Physicians | 5 (14.3) |
| Midwives | 19 (54.3) |
| Radiographers | 2(5.7) |
| Sonographers | 1(2.9) |
| Pharmacists | 3(8.6) |
| Nutritionists | 3(8.6) |
| Other | 1(2.9) |
| **Years of work experience (mean ± SD)** | 6.5 ± 4.1 |
| **Years of experience in facility (mean ± SD)** | 5.4 ± 4.5 |

## Participants experiences and perspectives

Participants shared numerous perspectives and experiences that can be categorized into four overarching themes: roles, facilitators, barriers and strategies to overcome the barriers. Embedded in these themes are sub-themes which provide in-depth information regarding the interprofessional team-based on respectful and dignified maternal care.

## A. Roles of healthcare professionals in the provision of respectful and dignified maternal care in team-based interprofessional settings

**Improve patient outcomes.** All healthcare professionals interviewed emphasized the improvement of patients' overall health outcomes, particularly, delivery of quality and tailored healthcare services, as their primary role in working within an interprofessional team setting. Participants highlighted the necessities of delivering comprehensive and high-quality healthcare services to enhance the health outcomes of mothers seeking maternal care at the facility. A physician in support of this assertion noted that, *"Personally, I think our primary role is providing quality healthcare to improve patients health outcomes.." (Physician, Male 04).* Healthcare professionals reiterated the importance for all cadres of professionals involved in patients' care at every stage, to be deeply passionate and strive for excellence in order to achieve optimal outcomes. A healthcare professional described:

*..and by this everybody should get to do whatever they're involved in at every stage....and we ought to be very passionate about the quality of care for mothers, we just have to give our best (Midwife, Female 05).*

**Enhance healthcare professional outcomes.** Most participants providing maternal care services within the interprofessional team-based setting mentioned that improving job performance and fostering team-based collaborative experiences was among their core duties as team members. They particularly remarked that as team members providing interprofessional team-based patient care, they essentially ensured optimal clinical efficiency and minimized errors during service delivery. A pharmacist observed, *"On numerous occasions as part of the team providing care at the wards, I have ensured that patients' drug prescriptions are accurate and effective, especially with antibiotics..Most inexperienced physicians can cause a mess...." (Pharmacist, Male 03)*

Healthcare professionals further mentioned that interprofessional team-based patient care encouraged professional autonomy as well as collaborative engagements in the delivery of respectful and dignified maternal care. Participants also echoed the sentiment that interprofessional team-based patient care enhances "job satisfaction" and minimizes healthcare professionals' "burnout and turnover intentions." A supportive quotation from a healthcare professional included:

*…it is my duty as a team member to ensure that clinical errors are reduced and I become comfortable and confident in my skill. I also have to improve my interpersonal skills needed to work with different cadres of health workers. Teamwork reduces the stress and tension on you and you achieve your goals (Midwife, Female 07).*

**Optimize facility outcomes.** Healthcare professionals' responses pointed to the role of interprofessional team-based maternal care team members in the enhancement of facility's organizational outcomes. Participants notably expressed that their role as team members within an interprofessional team-based maternal care setting enhanced collective ownership of the facility's values and goals. A nutritionist noted that for *"allied health professionals like myself, when we are part of the interprofessional team providing care we begin not only to deeply appreciate how things work within the clinical environment, we tend to appreciate and defend the facility's values and objectives."* (Nutritionist, Female 02).

Healthcare professionals also revealed that their role in interprofessional team-based maternal care culminates into reducing costs and expenses required to efficiently manage the maternal care facility. Most participants also echoed that their roles in interprofessional team-based maternal care assist in the management of healthcare professionals' ill-perceptions about their working environment, including leadership of facilities. One health professional stated that:

*When you are part of the diverse cadre of professionals providing care together at the wards you are privy to and part of discussion on a lot of management matters at the facility. In fact this changes certain ill-perceptions we had about the current working conditions. You realize that the leadership is trying their best (Midwife, Female 12)*

## B. Facilitators to the provision of respectful and dignified maternal care in team-based interprofessional settings

**Facility facilitators.** Participants' responses highlighted facilitators identified within facilities that enhance delivery of respectful and dignified maternal care within team-based interprofessional settings. According to participants, facilities rewarding team successes and team promoting behaviours enhance the delivery of respectful and dignified maternal care. A supportive comment shared by a healthcare professional included:

*…. I realized that incentives awarded to well-performing staff within the various teams of healthcare workers managing maternal wards motivated them to fully commit to clients' care… (Midwife, Female 09).*

They also reiterated that improving the work environment, particularly healthcare infrastructure within facilities also encouraged the delivery of interprofessional team-based maternal care. A physician remarked, *"...you see that refurbished MBU there, in its old state we lost a couple of babies and sometimes mothers periodically, due to absence and inade-quacy of consumables and appropriate technology. However, after it was rehabilitated and obsolete equipment replaced, we hardly hear of babies and maternal mortality of late…..."* (Physician, Female 01)

**Team-based facilitators.** Most of the healthcare professionals emphasized that team-based factors, particularly, role-clarity and open communication, plays a crucial role in facilitating the provision of respectful and dignified maternal care within team-based interprofessional settings. Participants further revealed that delivery of interprofessional respectful and dignified maternal care is enhanced when team members are aware of the outcomes of their collaborative efforts, thereby motivating them to perform their designated roles effectively. A healthcare worker mentioned that;

*...respectful care delivery is understood to be solely the responsibility of all team members, not a one-man show. Therefore, each individual is expected to fulfill their designated roles and responsibilities to achieve this care (Radiographer, Male 02).*

Most healthcare professionals echoed that honest, transparent and timely transmission of information among team members within an interprofessional setting enhances the delivery of respectful and dignified maternal care. A nurse said, *"..we need to communicate openly to ourselves very well and in simple terms not in medical jargons so that outcomes for mothers are optimal" (Nurse, Female 02)*

**Personal facilitators.** Healthcare professionals emphasized that individual traits of compassion and competence facilitates the delivery of respectful and dignified maternal care within team-based interprofessional settings. They mentioned that showing empathy, respect and concern for patients' wellbeing particularly enhanced respectful and dignified maternal care within team-based settings. A midwife in support of this assertion recounted, *"I wanted to be a midwife because I wanted to save babies and mothers. I lost my only sister through childbirth when I was young.." (Midwife, Female 05).*

Participants further highlighted that healthcare professionals' technical abilities, clinical expertise and application of up-to-date medical knowledge also improved the provision of respectful and dignified maternal care within team-based interprofessional facilities. A healthcare professional stated:

*Training and continuous professional development of health personnel is the key to providing respectful and dignified maternal care in this facility. Once the human resources are competent, even with the absence of infrastructure, he/she can improvise (Physician, Male 04)*

### C. Barriers to the provision of respectful and dignified maternal care in team-based interprofessional settings

**Facility barriers.** Healthcare professionals mentioned operational constraints, particularly, infrastructural limitations as barriers within healthcare facilities that act as impediments to the provision of respectful and dignified maternal care within team-based interprofessional settings. According to participants, infrastructural limitations, notably limited spaces within healthcare facilities, pose as obstacles in the provision of respectful and dignified maternal care. Healthcare professionals emphasized that due to limitations in spaces at the wards and consulting rooms, there is overcrowding which impedes respectful and dignified maternal care. A comment shared in support of this assertion included:

*The spaces in the wards are too small so during our rounds as a team, there is noise all over the place to the extent that team members sometimes are unable to appreciate whatever information is shared (Nutrition Officer, Female 01).*

They echoed that the inadequacy of privacy screens at the facility also resulted in poor adherence to patients' privacy and confidentiality. A midwife remarked, *"There are a lot of women occupying this small ward and because there are inadequate privacy screens sometimes it makes it difficult to ensure privacy and confidentiality" (Midwife, Female 01).*

**Team-based barriers.** Most participants emphasized that team-based factors, particularly, staff shortages, discrimination and disrespect impedes the delivery of respectful and dignified maternal care within team-based interprofessional settings. Participants revealed that poor staff strength stifles delivery of interprofessional respectful and dignified maternal care. A healthcare professional mentioned that;

*Currently, we are under-staffed so when you receive a lot of clients from the theatre, it becomes difficult to effectively monitor them..."(Midwife, Female 04).*

Participants echoed that discrimination and disrespect of certain cadres of healthcare professionals within an inter-professional team-based setting impacted negatively on the provision of respectful and dignified maternal care. A radiographer said, *"...you see, within such team-based care settings, contributions from allied health professionals are barely recognized." (Radiographer, Male 02)*

**Personal barriers.** Participants highlighted that negative attitudes and poor communication of healthcare professionals impede the provision of respectful and dignified maternal care within team-based interprofessional settings. Healthcare professionals revealed that negative attitudes, particularly anger, erodes delivery of respectful and dignified maternal care within team-based settings. A physician in support of this assertion recounted:

*Working on our skills and knowledge is essential, and equally important is addressing our attitudes, particularly anger issues, among various professionals, especially nurses. (Physician, Male 02).*

Participants further emphasized that poor interpersonal communication among healthcare professionals disrupts the delivery of respectful and dignified maternal care within team-based interprofessional settings. A participant recalled that, *"there should be proper communication among the team providing collaborative care to patients. For instance, once the labour ward has an issue, it has to be communicated for doctors, nurses, midwives and pharmacists among others to know (Midwife, Female 04).*

Healthcare professionals emphasized that negative attitudes among team members could deter patients from seeking timely healthcare services, posing as a significant barrier to the provision of respectful and dignified maternal care in interprofessional, team-based settings. One healthcare professional discussed that *"when pregnant women fail to seek timely maternal care as a result of the bad behavior of a midwife, it becomes difficult for the team at the maternity unit to detect and promptly deal with potential complications, especially when the women are brought in during labour." (Physician female 03)*

## D. Strategies to overcome the barriers

**Strengthening infrastructural inadequacies in facilities.** Participants emphasized that the infrastructural limitations in healthcare facilities need to be improved. They especially suggested that investment in the physical infrastructure (i.e., spaces and equipment) would help improve the provision of respectful and dignified maternal care within team-based interprofessional settings. For example, they mentioned that availability of spacious maternity wards was important because it will ensure that patients enjoy their privacy and confidentiality. Thus, they reiterated the urgent need for an expanded female ward. A supportive statement by a healthcare professional included:

*With spacious ward and consulting rooms, we will be able to serve our clients in dignity and confidence without worrying about spilling information… (Physician, Female 01)*

Most of the healthcare professionals also echoed similar sentiments emphasizing the provision of privacy screens at the wards and consulting rooms. Healthcare professionals also shared that addressing the limitation in privacy screens will enhance the respect of patients' right to privacy and confidentiality. Some healthcare professionals also suggested that the provision of well equipped mobile medical vans at specific open spaces across the metropolis to cater for emergency deliveries could augment the respectful and dignified maternal services provided at the facility. One healthcare professional remarked:

*..having an equipped mobile medical vans at open spaces across the metropolis can enhance safe emergency deliveries and also reduce the pressure on this facility (Midwife, Female 10)*

**Increasing healthcare staffing teams and their competence.** Most participants consistently expressed the need to increase the staff strength at the healthcare facility. They especially suggested that the recruitment of healthcare professionals (e.g., physicians and midwives) would assist improve delivery of respectful and dignified maternal care within team-based interprofessional settings. For instance, participants mentioned that the presence of an adequate number of physicians and midwives was important because their services were needed within the team-based interprofessional setting to deliver respectful and dignified maternal care at the hospital. Thus, they reiterated the urgent requirement for physicians and midwives. A statement in support of this assertion included:

*We usually shuffle and juggle among ourselves to provide services across the entire units of the hospital where our services are needed. It appears more hands may be required for efficiency (Physician, Female 01)*

Similarly, most healthcare professionals echoed remarks emphasizing the need for enhancing competence, particularly technical abilities, clinical expertise and application of up-to-date medical knowledge among healthcare professionals. They reiterated that such continuous professional development ensures healthcare professionals are abreast with the latest practices and knowledge, thereby enhancing the quality of care provided. Healthcare professionals further mentioned that continuous professional development is not just for the improvement of technical skills but also for better delivery of respectful and dignified maternal care within team-based interprofessional settings. Thus, healthcare professionals recommended that such continuous professional development should engage all cadres of staff involved in the provision of respectful and dignified maternal care within team-based interprofessional settings.

*..being aware of the ever-changing dynamics of medicine hence all team members are conscious of building our individual skills and clinical expertise to remain efficient in providing dignified care to patients (Physician, Male 04)*

**Enhancing attitudes of healthcare professionals.** As a strategy for enhancing access to respectful and dignified maternal care within team-based interprofessional settings, participants emphasized the need for changes in the negative attitudes, particularly, anger and disrespect among healthcare professionals. They recounted that in a team-based healthcare delivery environment, where the opinions of healthcare professionals were required in making treatment decisions, anger should be eschewed. A healthcare professionals remarked that:

*..discarding all unnecessary anger tendencies among team-based members within anIPCdelivery environment is crucial.. (Sonographer, Male 01)*

Most healthcare professionals also echoed similar views, reaffirming that participants could deliver respectful and dignified maternal care within team-based interprofessional settings when mutual respect is demonstrated. A healthcare professionals asserted:

*Team members can enhance respectful and dignified maternal care within team-based collaborative care environments when there is mutual respect no matter the background and rank of healthcare professionals… (Radiographer, Male 02)*

## Discussion

This study sought to investigate the perspectives and experiences of healthcare professionals on interprofessional team-based respectful and dignified maternal care in Ghana. Overall, the study highlights healthcare professionals' roles, facilitators, barriers and strategies to overcome barriers in a team-based interprofessional setting providing respectful and dignified maternal care in Ghana.

The various cadres of healthcare professionals play multifaceted roles in ensuring respectful and dignified maternal care within team-based settings. The primary roles identified in this study includes improving patient outcomes, enhancing healthcare professional outcomes, and optimizing facility outcomes. As participants in the study indicated, the improvement of patient outcomes involves providing high-quality, comprehensive services tailored to maternal needs, which ultimately enhance maternal and neonatal health [59,60]. Again, in enhancing healthcare professional outcomes there is the need to foster a collaborative team environment where errors are minimized and job satisfaction is improved, thus reducing burnout [61,62]. Additionally, optimizing facility outcomes through team-based care contributes to the effective utilization of resources and reduction of operational costs [63] as well as adherence to facility values [64].

Numerous facilitators contribute to dignified and respectful interprofessional team-based maternal care. Facilitators such as clear communication, role clarity, and shared goals identified in the study align with the literature asserting that these elements are critical in fostering effective interprofessional team-based maternal care [65]. Moreover, the importance of supportive infrastructure and rewards for teamwork, as emphasized by participants, highlights healthcare facilities' role in promoting interprofessional team-based maternal care [60]. Furthermore, personal traits such as compassion, empathy, and technical competence identified in the current study, are crucial for providing dignified and respectful maternal care, as they directly affect patient interactions and outcomes [16].

Despite the facilitators, several barriers impede the provision of dignified and respectful maternal care. Facility barriers such as inadequate infrastructure, limited space, and insufficient equipment for patients' privacy, as echoed by participants, can significantly hinder the delivery of care [4,7]. At the team level, challenges identified in the current study such as unclear roles, poor communication, and staff shortages often lead to fragmented and inconsistent care [6,7]. Also, the study's findings at the personal level, including healthcare professionals' attitudes, such as resistance to collaborative practices and poor interpersonal skills, can minimize interprofessional team-based maternal care effectiveness and patient care quality [66]. Addressing these barriers requires organizational adjustments that go beyond the superficial changes in workflow or policy, emphasizing a transformative approach to healthcare delivery systems [67].

In addressing these barriers to team-based interprofessional respectful and dignified maternal care, there is the need to adopt multifaceted strategies. The strategies proposed by participants, such as improving infrastructural facilities, enhancing staff competence, and fostering positive attitudes, are essential for overcoming challenges with team-based interprofessional respectful and dignified maternal care. The findings in the current study are consistent with the literature. Infrastructure improvements are essential to provide adequate space and privacy as prescribed by the RMC charter [36], which are fundamental to respectful care [48]. Increasing staff numbers and enhancing their competencies through continuous professional development are vital for maintaining a high standard of care and fostering positive team dynamics [47]. Furthermore, other scholars propose promoting an IPE framework that includes training in collaboration and communication to significantly improve team functioning and care coordination [68].

Integrating the roles, facilitators, barriers, and strategies into the provision of respectful and dignified maternal care within team-based interprofessional settings necessitates a concerted effort from all stakeholders in healthcare provision. Facilities must support healthcare professionals working in team-based interprofessional settings by fostering a conducive culture, continuous improvement in skills and attitudes, and enhance policies to facilitate the structural conditions necessary for effective interprofessional teamwork. By addressing these elements, healthcare settings can improve respectful and dignified maternal outcomes, enhance job satisfaction among healthcare professionals, and optimize operational efficiency, thereby creating a sustainable model for delivering high-quality maternal care [65,69]. This holistic approach not only supports the well-being of the mother and child but also promotes a resilient healthcare system capable of adapting to the complexities of maternal health needs.

## Limitations

There are limitations to this study, notwithstanding its valuable insights into interprofessional team-based maternal care in Ghana. First, the adoption of qualitative methodologies, although useful for harnessing in-depth understanding, there is a limitation to generalizing the study's outcomes. This is so because the experiences and perspectives of healthcare professionals recruited for this study are contextual and may not represent the opinions at all settings within Ghana or other regions with similar healthcare environments. Additionally, there are potential biases inherent in self-reported data, where participants might provide socially desirable responses or omit negative experiences due to their familiarity or otherwise with interviewers.

## Conclusion

The qualitative study highlights the critical importance of interprofessional team-based maternal care in Ghana ensuring respectful and dignified treatment of mothers. The multifaceted roles of healthcare professionals in improving patient outcomes, enhancing their own job satisfaction, and optimizing facilities' operations aimed at enhanced respectful and dignified maternal care were highlighted. Key facilitators identified included rewarding team successes, effective communication, role clarity, compassion and individual competencies. Conversely, barriers such as poor infrastructure, insufficient staffing and negative attitudes of healthcare professionals significantly impeded the delivery of respectful and dignified maternal care. A multisectoral approach to enhanced respectful and dignified maternal care within team-based interprofessional settings included strengthening facility's infrastructure, increasing the workforce and building the capacity of healthcare professionals through continuous professional development. Consequently, the study revealed how interprofessional care fosters respectful and dignified maternal services through improving organizational cohesion, professional autonomy, job satisfaction, and positive perceptions of healthcare leadership, working conditions, and resource allocation. This suggests that multistakeholder solutions do not only promise delivery of respectful and dignified maternal care but also supports a more integrated and efficient healthcare system, especially required in low- and middle-income countries (LMICs).

## Study implications

The study's findings imply an urgent need for holistic strategies to strengthen interprofessional collaboration essential for respectful and dignified maternal care. It is obvious that legislative, economic, organizational, educative and research interventions are crucial to improving respectful and dignified maternal care at the study setting [70]. In terms of legislative interventions, Ghana and other relevant LMICs should enact and enforce laws addressing abuse (verbal and physical) of mothers seeking healthcare services at all levels, focusing on respect, dignity, informed consent, privacy and confidentiality. Similarly, government through the health ministry should identify and prioritize investments in healthcare infrastructure and healthcare staffing in order to close the existing healthcare availability gaps. In terms of organizational interventions, authorities within the health sector should adopt and institutionalize globally accepted and locally viable interprofessional approaches to the planning, design, delivery, and evaluation of maternity care in Ghana. Additionally, regular monitoring, supervision and certification of local maternity units should align with international standards. Educational interventions aimed at integrating human rights principles of maternal care within interprofessional training and professional development curricula for medical, nursing, midwifery, health administration and other relevant officers should be mandatory. For instance, healthcare training authorities could draw from relevant existing frameworks such as the RMC charter. Grounded in international and regional human rights laws, the RMC charter articulates the dignity, safety, and respect for women and newborns during maternal care [36]. Moreover, policies that foster collaborative practices and IPE remain crucial. Relative to research interventions, robust, multidisciplinary, and cross-national studies are critical to understanding and addressing respectful and dignified maternal care within team-based interprofessional settings. Such research should guide decision-making processes for policymakers, healthcare providers, and families, enabling them to collaboratively address systemic

barriers to respectful and dignified maternal care. These strategies will contribute to shaping policy and practice, aimed at improving the structural environment necessary for effective interprofessional teamwork, thereby enhancing high-quality, respectful and dignified maternal care in Ghana and similar contexts.

## Acknowledgments

We are grateful to all participants for their deep insights and time spent in sharing their experiences.

## Author contributions

**Conceptualization:** Veronica Millicent Dzomeku, Ebenezer Dassah, Ebenezer Mensah Gyimah, Abigael Omowumi Emikpe, Lydia Boampong Owusu, Catherine Kroamah Dwumfour, Oluwatoyin Abayomi Ogunyewo, Thomas Peprah Agyekum, Eric Adjei Boadu, Emmanuel Kweku Nakua.

**Data curation:** Veronica Millicent Dzomeku, Ebenezer Dassah, Ebenezer Mensah Gyimah, Abigael Omowumi Emikpe, Lydia Boampong Owusu, Catherine Kroamah Dwumfour, Oluwatoyin Abayomi Ogunyewo, Thomas Peprah Agyekum, Eric Adjei Boadu, Emmanuel Kweku Nakua.

**Formal analysis:** Veronica Millicent Dzomeku, Ebenezer Dassah, Ebenezer Mensah Gyimah, Catherine Kroamah Dwumfour, Emmanuel Kweku Nakua.

**Funding acquisition:** Veronica Millicent Dzomeku.

**Methodology:** Veronica Millicent Dzomeku, Ebenezer Dassah, Ebenezer Mensah Gyimah, Abigael Omowumi Emikpe, Lydia Boampong Owusu, Catherine Kroamah Dwumfour, Oluwatoyin Abayomi Ogunyewo, Thomas Peprah Agyekum, Eric Adjei Boadu, Emmanuel Kweku Nakua.

**Project administration:** Veronica Millicent Dzomeku.

**Resources:** Ebenezer Dassah, Abigael Omowumi Emikpe.

**Software:** Ebenezer Dassah, Ebenezer Mensah Gyimah.

**Supervision:** Veronica Millicent Dzomeku, Ebenezer Dassah, Oluwatoyin Abayomi Ogunyewo, Emmanuel Kweku Nakua.

**Validation:** Veronica Millicent Dzomeku, Ebenezer Dassah, Ebenezer Mensah Gyimah, Abigael Omowumi Emikpe, Catherine Kroamah Dwumfour, Oluwatoyin Abayomi Ogunyewo, Thomas Peprah Agyekum, Emmanuel Kweku Nakua.

**Visualization:** Veronica Millicent Dzomeku, Lydia Boampong Owusu, Oluwatoyin Abayomi Ogunyewo.

**Writing – original draft:** Veronica Millicent Dzomeku, Ebenezer Dassah, Abigael Omowumi Emikpe, Lydia Boampong Owusu, Catherine Kroamah Dwumfour, Thomas Peprah Agyekum, Eric Adjei Boadu, Emmanuel Kweku Nakua.

**Writing – review & editing:** Veronica Millicent Dzomeku, Ebenezer Dassah, Ebenezer Mensah Gyimah, Abigael Omowumi Emikpe, Lydia Boampong Owusu, Catherine Kroamah Dwumfour, Thomas Peprah Agyekum, Eric Adjei Boadu, Emmanuel Kweku Nakua.

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
