## [Decision Letter · Decision Letter 0]

9 Dec 2024

PGPH-D-24-02407

Interprofessional team-based respectful and dignified maternal care in Ghana: A qualitative study

Dear Dr. Nakua,

Thank you for submitting your manuscript to PLOS Global Public Health. After careful consideration, we feel that it has merit but does not fully meet PLOS Global Public Health’s publication criteria as it currently stands. Therefore, we invite you to submit a revised version of the manuscript that addresses the points raised during the review process.

Thank you for this very relevant and contextual qualitative study on respectful and dignified maternal care in Ghana. Please be sure to apply all changes suggested by the reviewers for acceptance. No conflict exists between the reviews and authors should kindly follow the guidance for minor reviews of both. The distinction should be made between the terms interdisciplinary and inter-professional, as there is a fundamental pedagogic difference between the terms.

In addition to the reviewer's comments, do include a positionality statement for the authors in the text as a sheading in the methods. Kindly add in the Consolidated criteria for reporting qualitative research (COREQ) checklist or the Standards for reporting qualitative research: a synthesis of recommendations *as an appendix * and ensure that the requirements for qualitative research are clearly identified from the study.

To reiterate one reviewer's observation, it is important to really think through the language of the manuscript and highlight the concepts of *respectful maternity care * as engaged in literature with a stronger human rights lens.

We look forward to receiving your revised manuscript.

Kind regards,

Barnabas Tobi Alayande

Academic Editor

Journal Requirements:

Additional Editor Comments (if provided):

Reviewers' comments:

Reviewer's Responses to Questions

**Comments to the Author**

1. Does this manuscript meet PLOS Global Public Health’s publication criteria ? Is the manuscript technically sound, and do the data support the conclusions? The manuscript must describe methodologically and ethically rigorous research with conclusions that are appropriately drawn based on the data presented.

Reviewer #1: Yes

Reviewer #2: Yes

2. Has the statistical analysis been performed appropriately and rigorously?

Reviewer #1: Yes

Reviewer #2: Yes

3. Have the authors made all data underlying the findings in their manuscript fully available (please refer to the Data Availability Statement at the start of the manuscript PDF file)?

Reviewer #1: Yes

Reviewer #2: Yes

4. Is the manuscript presented in an intelligible fashion and written in standard English?

Reviewer #1: Yes

Reviewer #2: Yes

5. Review Comments to the Author

Reviewer #1: 1. The research methodology is clear with evidence of steps taken to enhance trust worthiness. The findings of this study are linked to the conclusion.

2. Data adequately analysed. Themes were clearly identified and findings reported under each theme.

3. Author has stated that all data is available at the Committee on Human Research, Publications and Ethics at the Kwame Nkrumah University of Science and Technology, Kumasi. Ghana.

4. Manuscript is written in standard English.

Please find below few comments for your consideration.

1. Some literature on disrespectful experiences of mothers seeking maternity care in Ghana in the introduction would be helpful in justifying the need for this study.

2. Any new knowledge this study adds to literature should clearly be captured in the conclusion.

3. An Omission identified in line 26 '' Of health care providers in a collaborative team-based setting at a health ''FACILITY'' in Ghana.

4. omission in line 473 ''maternal care in Ghana '' IN'' ensuring ....

Reviewer #2: Overall, this paper is excellent, timely and crucial in light of the increased advocacy happening around human rights in childbirth. It is well written and adopts a unique approach, i.e. interprofessional team-based approach to improving the quality of care.

The abstract: Authors may want to strengthen the abstract by structuring it in a way that clearly shows that the paper looks at how interprofessional team-based care is supposed to contribute to respectful and dignified care. When reading the abstract, the connection between these two themes is separated as more emphasis is laid on interprofessional team-based care alone and not on how interprofessional team-based care is supposed to lead to respectful and dignified care.

Lines 25 and 26 feel a bit incomplete.

51 and 52 need restructuring to avoid repetition.

54, meanwhile, may not be the most appropriate transition phrase for that sentence

You make use of both these terms: Interdisciplinary vs interprofessional- Not sure whether they are meant to refer to the same thing. Consistency is important

In the conclusion and recommendation section, the authors may need to be a bit more clear with regard to who is being called upon to address the structural barriers. While it is not wrong to ask the facilities to address structural issues, the problem runs deeper than that. Most structural issues affecting service provision fall within the state’s duty to promote, protect and fulfil the right to health. Placing this responsibility on facilities may be a bit misleading. The same applies to issues of hospital staffing and continuous professional development. Who exactly is responsible for this between the state and the facility? see Sadler, M., Santos, M. J., Ruiz-Berdún, D., Rojas, G. L., Skoko, E., Gillen, P., & Clausen, J. A. (2016). Moving beyond disrespect and abuse: addressing the structural dimensions of obstetric violence. Reproductive health matters, 24(47), 47–55. https://doi.org/10.1016/j.rhm.2016.04.002;
https://www.ohchr.org/sites/default/files/Documents/Publications/Factsheet31.pdf

While the authors adopt the language of respectful and dignified care, they may want to familiarise themselves with the language of Respectful Maternity Care. This may be a strategic approach to strengthen this body of work further as it provides an avenue to adopt a rights-based framing for certain issues raised in the paper. For example, the authors raise concerns about privacy and confidentiality, which are human rights issues that have been discussed in the Respectful Maternity Care Charter by the White Ribbon Alliance. To be truly holistic, an approach to improving women's experience during pregnancy, childbirth, and the postpartum period must Incorporate a rights-based lens. The spirit of RMC, which authors call respectful and dignified care, is about centring the birthing person’s human rights in providing care.

6. PLOS authors have the option to publish the peer review history of their article (what does this mean? ). If published, this will include your full peer review and any attached files.

**Do you want your identity to be public for this peer review?** For information about this choice, including consent withdrawal, please see our Privacy Policy .

Reviewer #1: **Yes: ** Irene Devine Dzirasa

Reviewer #2: No

---

## [Decision Letter · Decision Letter 1]

5 Feb 2025

Providers perspectives on a team-based maternal health care delivery in Ghana: A qualitative study

PGPH-D-24-02407R1

Dear Dr Nakua,

We are pleased to inform you that your manuscript 'Providers perspectives on a team-based maternal health care delivery in Ghana: A qualitative study' has been provisionally accepted for publication in PLOS Global Public Health.

Congratulations and best regards,

Ashti Doobay-Persaud

Academic Editor

Reviewer Comments (if any, and for reference):

Reviewer's Responses to Questions

**Comments to the Author**

1. If the authors have adequately addressed your comments raised in a previous round of review and you feel that this manuscript is now acceptable for publication, you may indicate that here to bypass the “Comments to the Author” section, enter your conflict of interest statement in the “Confidential to Editor” section, and submit your "Accept" recommendation.

Reviewer #1: All comments have been addressed

Reviewer #2: All comments have been addressed

2. Does this manuscript meet PLOS Global Public Health’s publication criteria ? Is the manuscript technically sound, and do the data support the conclusions? The manuscript must describe methodologically and ethically rigorous research with conclusions that are appropriately drawn based on the data presented.

Reviewer #1: Yes

Reviewer #2: Yes

3. Has the statistical analysis been performed appropriately and rigorously?

Reviewer #1: Yes

Reviewer #2: Yes

4. Have the authors made all data underlying the findings in their manuscript fully available (please refer to the Data Availability Statement at the start of the manuscript PDF file)?

Reviewer #1: Yes

Reviewer #2: Yes

5. Is the manuscript presented in an intelligible fashion and written in standard English?

Reviewer #1: Yes

Reviewer #2: Yes

6. Review Comments to the Author

Reviewer #1: Author has addressed comments from previous review.

Manuscript meets PLOS Global Public Health Criteria.

Research methodology is clear with steps taken to ensure trustworthiness.

Reviewer #2: (No Response)

7. PLOS authors have the option to publish the peer review history of their article (what does this mean? ). If published, this will include your full peer review and any attached files.

**Do you want your identity to be public for this peer review?** For information about this choice, including consent withdrawal, please see our Privacy Policy .

Reviewer #1: **Yes: ** Irene Devine Dzirasa

Reviewer #2: No
